# The usage of a modified straight-leg raise neurodynamic test and hamstring flexibility for diagnosis of non-specific low back pain: A cross-sectional study

Joan Hermosura[1]*, Everett Lohman, III[1°], Brenda Bartnik-Olson[2°], Jonathan Venezia[3°], Noha Daher[4°]

1 Department of Physical Therapy, School of Allied Health Professions, Loma Linda University, Loma Linda, California, United States of America, 2 Department of Radiology, Loma Linda Medical Center, Loma Linda University, Loma Linda, California, United States of American, 3 Department of Otolaryngology and Head/Neck Surgery, Loma Linda Medical Center, Loma Linda University, Loma Linda, California, United States of America, 4 Allied Health Science, School of Allied Health Profession, Loma Linda University, Loma Linda, California, United States of America

☉ These authors contributed equally to this work.
* jhermosura@students.llu.edu

**Data Availability Statement:** All relevant data are within the paper.

## Abstract

### Objectives

The main purpose of this research study was to compare mean modified straight-leg raise test (mSLR) and hamstring muscle length (HL) between chronic non-specific low back pain (LBP) and healthy subjects to understand the possibility of neuropathic causes in LBP population as it may impact the diagnosis and treatment of LBP. Another purpose was to compare mean mSLR between those with lumbar nerve root impingement and those without as determine by magnetic resonance imaging (MRI).

### Methods

The design of the study is cross sectional and included 32 subjects with ages ranging from 18–50 years old. Clinical exam objective measures were collected such as patient questionnaires, somatosensory tests, HL range of motion, and a mSLR test, and were compared to the findings from a structural lumbar spine MRI.

### Results

There were no significant differences in mean HL angulation and mSLR angulation between LBP and healthy subjects (p>0.05). There was no significant difference in mean HL by impingement by versus no impingement (38.3±15.6 versus 44.8±9.4, p = 0.08, Cohen's d = 0.50). On the other hand, there was a significant difference in mean mSLR angulation by impingement (57.6.3±8.7 versus 63.8±11.6, p = 0.05, Cohen's d = 0.60).

**Funding:** The Financial Disclosure statement should be the following: "This study was approved by the Institutional Review Board of Loma Linda University, Loma Linda, California (IRB# 5200311). Dr. Everett Lohmann, a contributing author, received a grant that supported the study. The School of Allied Health Professions SEED grant (STAR20067) was funded and endorsed by Dr. Craig Jackson. The funder, Dr. Craig Jackson, had no role in study design, data collection and analysis, decision to publish, or preparation of the manuscript.

**Competing interests:** The authors have declared no competing interests exist.

## Conclusions

The mSLR test was found to be associated with lumbar nerve root compression, regardless of the existence of radiating leg symptoms, and showed no association solely with the report of LBP. The findings highlight the diagnostic dilemma facing clinicians in patients with chronic nonspecific LBP with uncorrelated neuroanatomical image findings. Clinically, it may be necessary to reevaluate the common practice of exclusively using the mSLR test for patients with leg symptoms. This study may impact the way chronic LBP and neuropathic symptoms are diagnosed, potentially improving treatment methods, reducing persistent symptoms, and ultimately improving disabling effects.

## Introduction

Chronic musculoskeletal conditions, such as neck and back pain, are a global burden [1] and are among top reasons patients seek medical attention [2, 3]. Those with musculoskeletal conditions have a two-fold chance of developing a chronic disease of other body systems [4]. Chronic low back pain (LBP) is the leading cause of disability preventing engagement in work and daily activities [5]. From 1990–2015, a 54% increase in disability related to LBP was noted [6, 7]. About 70–80% of people will experience LBP in their lifetime [8, 9], with a linear increase in prevalence beyond their 3<sup>rd</sup> decade of life. Twenty percent of those with reported acute and sub-acute back pain will become chronic in nature [10]. Of those diagnosed with chronic back pain, 85% are labeled as non-specific- without an attributable anatomical cause for the symptoms [11].

Chronic LBP is complex and may be difficult to diagnose because of its multifactorial nature, thus making it difficult to determine specific causes. Acknowledging the complexity of chronic pain conditions, which expands beyond the basic criteria of symptoms lasting more than three months, the International Association for the Study of Pain (IASP) categorized chronic pain into primary and secondary syndromes [12]. Chronic primary pain is defined as pain in one or more anatomical regions that, persists or recurs for longer than 3 months, is associated with significant emotional distress and/or functional disability that affects daily roles and activities, with symptoms that are not better accounted for by another diagnosis [13]. A classification system helps healthcare providers understand underlying mechanism and characteristics of chronic non-specific LBP that may not be obvious from pathoanatomical contributors.

Common practice for diagnosing LBP and related symptoms includes a subjective exam, investigating patient's medical history, a potential neurological screen, and typically an imaging modality like a Magnetic resonance image (MRI). Effectiveness of imaging is limited when no identifiable lesion or disease can be attributed to symptoms. In terms of lumbar spine radiculopathy, there is weak association between MRI findings and symptom presentation [14]. Similarly, abnormal pathoanatomical findings may be seen in asymptomatic people [15, 16]. Treatment difficulty and prevalence of chronic LBP may be related to misdiagnosis stemming from the poor association between MRI findings (or lack thereof), varied symptom descriptors, and the interpretation of those descriptors.

Medical doctors on average spend about 5 minutes with patients [17], placing emphasis on subjective nature of symptoms. Researchers highlight challenges with subjective pain descriptions, complicating clinicians' diagnostic assessment [18]. Common nerve pain descriptors are also used to describe chronic musculoskeletal concerns such as chronic neck pain, chronic

back pain, headaches, and osteoarthritis. Symptoms attributed to a neuropathic etiology generally follow dermatome and myotome patterns yet, similar patterns and sensory changes are seen in musculoskeletal conditions [18–29].

Recognizing neuropathic influences in chronic musculoskeletal conditions is important for effective diagnosis and treatment, contributing to a reduction in chronicity and disability. While symptoms in chronic musculoskeletal and neuropathic conditions may overlap, the evaluation and treatment are very different. If symptoms are assumed to have a neurological contribution, clinically the performance of a neurological exam and a neurodynamic test is typical. Specific to the low back, this clustering of tests may include a lower extremity neurodynamic test- such as the modified straight leg raise (mSLR) test. The mSLR test includes passive movement of the ankle into full dorsiflexion while maintaining the knee in extension, followed by passive hip flexion. In the presence of neural inflammation, nerve fibers become sensitive to stretch with low pressure and minimal tension [30]. Mechanosensitivity to stretch provokes the lumbosacral plexus and those peripheral nerves distal to it. This mechanism is thought to be the pain generator during the mSLR maneuver [31, 32]. When spinal cord or nerve root compression is not visible on MRI, a positive mSLR test may indicate an evaluation of nerve mechanosensitivity rather than diagnostic compressive syndromes [33]. Despite the prevalence of back and leg pain, differing opinions on diagnostic procedures and their efficacy creates challenges in diagnosing and treating cases without a visible pathoanatomical root cause.

The current evaluation process is limited in terms of precision in identifying causes of LBP symptoms. Recognizing the potential role of neuropathic causes in chronic musculoskeletal conditions may impact the diagnosis and treatment of LBP. The research aimed to investigate potential neuropathic causes in chronic non-specific LBP by comparing mean mSLR and HL between those with LBP and healthy subjects. Additionally, the study sought to assess the significance of mean mSLR difference in those with lumbar impingement, as determined by MRI, compared to those without. It was hypothesized that the mSLR angulation will be lower in subjects with LBP regardless of radiating leg symptoms, that the mSLR angulation will be associated with lumbar nerve root impingement, and that there is an interaction between LBP, lumbar nerve root impingement, and the mSLR.

## Methods

This study was approved by the Institutional Review Board of Loma Linda University, Loma Linda, California (IRB# 5200311) on 09/18/2020 and was completed in 07/2022. All subjects participated voluntarily and provided their written informed consent to participate in the study. Recruitment of participants began upon approval on 9/18/2020 and terminated on 7/29/2022. Due to the global pandemic of COVID-19, research data collection at Loma Linda University was paused leading to a lengthened participant recruitment period. Participants were recruited by word of mouth, flyers, emails, and Loma Linda University approved social media. A total of 32 subjects were recruited. Back pain (or lack thereof) was solely self-reported and accepted at face value. Inclusion criteria for the Healthy Group (no LBP) were: (a) healthy adults ages18-50 years old, (b) No symptoms of back pain, related leg pain, and/or radicular pain within the last year or currently, (c) functional hip flexion ROM, knee extension range of motion (ROM), and ankle ROM. Participants experiencing back pain consistently for >3months were considered to have chronic symptoms. Specifically, inclusion criteria for the Back Pain Group(LBP) were: (a) adults ages18-50 years old, (b) functional hip flexion ROM, knee extension ROM, and ankle ROM, (c) current or history of LBP symptoms greater than 3 months, (d) low back region defined as pain located in the inferior/posterior rib angle to gluteal crease (e) willingness to pause current usage of opioids, anti-depressants, and/or other

nerve related medication 3 hours prior to start of study. After meeting inclusion criteria, the back-pain group was divided into two sub-groups: "impingement" and "no impingement" based on MRI findings.

Exclusion criteria were: (a) chronic pain syndromes or disorder such as fibromyalgia, chronic regional pain syndrome, irritable bowel syndrome, (b) connective tissue disorders such as Ehlers Danlos Syndrome (EDS), Rheumatoid Arthritis (RA), polymyositis, Sjogren's syndrome, (c) history of any spinal or lower extremity surgery/injury resulting in confirmed nerve damage, (d) current lower extremity pain without chronic LBP, and/or (e) presence of MRI contraindications such as implanted device, metallic foreign body, self-reported pregnancy, claustrophobia, etc.

## Design and setting

The design of the study was cross-sectional. The study was conducted at Loma Linda University, California where subjects completed 3 phases of the study after informed consent and protected health information (PHI) authorization: phase 1) subjective demographics and symptom questionnaires, phase 2) somatosensory and range of motion assessment, and phase 3) structural MRI of lumbar spine to determine the presence or absence of impingement.

In the first phase, subjects completed three questionnaires related to demographics, disability, and quality of symptoms. Details on the questionnaires are described in 'Phase 1: subjective demographics and questionnaires section' (see below). During the second phase subjects completed a battery of somatosensory tests based on the Qualitative Sensory Test (QST), which included mechanical pain threshold using pinprick stimulus, mechanical detection threshold assessment using Von-Frey filaments, vibration detection threshold assessment using 128 Hz tuning fork, and pain pressure threshold (PPT) assessment using a pressure algometer (J Tech Medical Industries, Salt Lake City, UT). Further details on the performance of each assessment are described in the 'Phase 2: somatosensory and range of motion' section (see below). Regions assessed included the low back, thigh (upper leg), calf region (lower leg), and the foot. Sites chosen are common lumbar spine neurological pain referral regions. Subjects were then screened for functional and pain free lower extremity range of motion. All screening physical examination procedures were performed by a licensed Doctor of Physical Therapy with nine years of clinical experience in outpatient orthopedics, including residency and fellowship training (JRH).

**Phase 1: subjective demographics and questionnaires.** *Oswestry disability index.* The Oswestry Disability Index (ODI) is a widely used pen-paper self-report questionnaire that is designed to assess the level of disability experienced by individuals with LBP. The ODI measure consists of ten items, each of which relates to a different aspect of daily functioning, such as personal care, lifting, walking, and sitting. For each item, the subject rates the level of difficulty experienced on a six-point scale ranging from "no difficulty" to "impossible". The scores for each item are summed to give a total score out of 50, with higher scores indicating greater disability. The ODI is a reliable and valid measure of disability [34]. The following are the categories used to classify the level of disability based on the ODI score [35]:

- 0% - 20%: Minimal disability

- 21% - 40%: Moderate disability

- 41% - 60%: Severe disability

- 61% - 80%: Crippled

- 81% - 100%: Bed-bound.

*PainDetect questionnaire.* The PainDetect Questionnaire (PD-Q) is a self-administered questionnaire designed to screen for neuropathic pain in individuals with chronic pain. The PD-Q consists of nine items, seven of which are related to sensory symptoms such as burning, tingling, and numbness, and two of which are related to pain quality and intensity. The subject rates the frequency and intensity of these symptoms on a five-point scale, with higher scores indicating more severe symptoms [6]. The PD-Q has high sensitivity and specificity in detecting neuropathic pain and is widely used as a screening tool for neuropathic pain in various chronic pain conditions [6]. The total score on the PD-Q ranges from 0 to 38, with higher scores indicating a higher likelihood of neuropathic pain. Based on the total score, the likelihood of neuropathic pain can be classified as follows [6]:

- Scores 0 to 12: unlikely to be neuropathic pain

- Scores 13 to 18: possible neuropathic pain

- Scores 19 to 38: likely neuropathic pain

**Phase 2: somatosensory and range of motion assessment.** *Mechanical detection threshold assessment.* Mechanical detection threshold assessment using Von-Frey Hairs (filaments) or Semmes-Weinstein Monofilaments are commonly used in sensory testing to assess the threshold of pressure or touch sensation in patients with neurological or musculoskeletal disorders [36]. They are considered reliable and valid instruments for detecting subtle changes in sensory function over time and are often used in research studies to assess the effectiveness of various treatments and interventions. Specifically, the 10-gram monofilament is an indicator for loss of protective touch specifically for peripheral neuropathy. Subjects were supine with legs extended and instructed to describe the location felt if sensed. Five regions representative of lumbar segments were chosen bilaterally (anterior thigh, medial femoral condyle, medial malleoli, webspace of great toe, and Achilles region). Monofilament was pressed on to the skin surface at a 90-degree angle until filament bowed and was held for 1–2 seconds while waiting for the subject to respond.

*Mechanical pain threshold assessment.* Mechanical pain threshold assessment is part of a battery of somatosensory tests in QST [37] to assess sensory A-delta fibers. Specifically, these fibers are important for somatosensory detection related to protective pain and temperature sensing ability. For clinical bed side examination, a wooden toothpick or weighted monofilament may be utilized [38, 39]. To assess the mechanical pain threshold a sharp sensation provided by a calibrated Neurotip® device (Neuropen ®, Owen Mumford, England) was used. The calibrated Neurotip was used to maintain consistency in the amount of force used during application; depression of the tip was applied to a pre-established 40g marker on the Neuropen®. Subjects were supine with legs extended and instructed to describe the location felt if sensed. The sharp sensation was familiarized to the subject with a trial on the posterior hand. The Neurotip was pressed on to the skin surface at a 90-degree angle; the tip was applied until the 40g marker was reached, and then held on the skin for 1–2 seconds. Clinician then waited for subject to respond.

*Pain pressure threshold assessment.* The Pain Pressure Threshold Assessment (PPT) assessment was pulled from a standardized battery of tests to quantify somatosensation, specifically for deep pain sensitivity mediated by muscle C- and A-delta fibers [37]. The test measured the amount of pressure stimulus tolerated resulting in a painful pressure sensation. Individuals with nerve compromise may have an altered tolerance to pressure. The PPT assessment was performed using an electronic digital pressure algometer device (Algometer [TM] Commander, J-Tech Medical Industries, USA) with a probe of 1cm$^2$ rubber end to assess and quantify

mechanical sensitivity. The PPT assessment is a low-cost and time efficient method to determine somatosensory dysfunction [40–42]. Subjects were supine with legs extended and a clicker was given to press notifying the tester when to stop recording pressure. Subjects were instructed to press the clicker when sensation immediately changed from "pressure to pain." The clinician held the probe in the right hand and applied the probe perpendicular to the location of interest [33]. To determine segmental sensitivity in a chronic LBP population, similar methods incorporated by [43] were performed. Five regions representative of lumbar segments were chosen bilaterally (lumbar paraspinals, middle quadricep, tibialis anterior, medial instep, and medial superior gastrocnemius region). Once the subject pressed the clicker, mechanical pressure stopped, the device was removed, and data was electronically recorded. Three trials of PPT assessment were performed and the average measurement was calculated and recorded.

*Vibration detection threshold assessment*. Vibration detection threshold assessment was performed using a traditional, clang 128-Hz tuning fork. A 128-hz tuning fork is commonly used to assess vibration perception in patients with neuropathic conditions [38, 44, 45]. The tuning fork is cost effective alternative to expensive vibration sensation testing [46, 47], which has moderate inter-rater reliability and a positive likelihood to predict neuropathy [48]. A timed assessment method was used which measures the maximal time (in seconds) needed for vibration stimulus to become undetectable [37, 46, 49]. The subject was placed supine and given a clicker with instructions to immediately press when vibration stimulus was no longer detected. Subjects were familiarized with the vibration sense in the upper extremity before beginning. The clinician activated the tuning fork and then placed on 4 various regions bilaterally- medial and lateral malleoli and medial and lateral femoral epicondyle.

*Unilateral popliteal angle test*. Hamstring length was measured with a goniometer during the Unilateral Popliteal Angle (UPA) test also referred to as the Popliteal Angle Test (PAT). Hamstring length is may alter alignment and mechanics of the lumbopelvic region. Subjects were supine, and with the contralateral leg straight on the plinth and the test hip maintained in 90-degrees of flexion, the clinician passively extended the knee until the point where resistance was encountered indicating functional hamstring tightness [50]. Angulation closer to 0 degrees was indicative of hamstring flexibility.

*Modified Straight Leg Raise (mSLR) test*. The mSLR test, a neurodynamic test, was used to assess the mobility and mechanosensitivity of neural tissue. The concept of mechanosensitivity, is used in musculoskeletal and neurological assessments to describe how sensitive a tissue or nerve is to mechanical stress or pressure. An inclinometer was used to measure the hip joint range of motion at the end position of the mSLR test [51]. This technique is a valid and reliable method for measuring hip joint range of motion and can provide important information about neural tissue mobility and mechanosensitivity [51]. [52] investigated the reliability of measuring hip joint range of motion during the mSLR test and found good intra-rater reliability (ICC = 0.92) and inter-rater reliability (ICC = 0.90) for the inclinometer measurements. Another study by [53] investigated the correlation between the inclinometer measurements of hip joint range of motion during the mSLR test and the radiographic measurements of lumbar lordosis angle in patients with LBP. The study found a significant correlation between the inclinometer measurements and the radiographic measurements, indicating that the inclinometer measurements are an accurate reflection of hip joint range of motion during the mSLR test. A systematic review by [54] determined that inclinometer measurements had good to excellent reliability and validity for measuring hip joint range of motion during the mSLR test in healthy subjects and patients with LBP. To maximally tension the lower extremity neural tissue, the mSLR test involved the passive elevation of the leg with the knee maintained in extension and ankle in dorsiflexion; the hip is flexed towards 90 degrees with the subject in supine and the contralateral lower limb extended. The clinician continued with leg elevation until the

subject experienced pain, reproduction of symptoms, or other abnormal sensations in the lower limb or lumbopelvic region during the test or until tissue resistance was encountered just prior to pelvic rotation. Sensitizing-desensitizing maneuvers (hip adduction/abduction or dorsiflexion/plantar flexion) were performed based on location of reported pain or resistance to confirm peripheral nerve involvement.

**Phase 3: structural MRI.** The third phase of the study included MRI imaging of the lumbar spine; a commonly used reference standard for the diagnosis of suspected lumbar pain with/without sciatica. A 3T Siemens SKYRA system (Siemens Medical Solutions, Erlangen, Germany) was used to collect T2 weighted axial and sagittal images to determine presence/absence of nerve impingement for all subjects. MRI images were reviewed by an American Board of Radiology certified neuroradiologist with 9 years of experience (EP) who was blinded to all subject's demographics and characteristics. Nerve impingement grouping (impingement" or "no impingement") was determined by the neuroradiologist based on a combination of the integrity of the lumbar anatomical regions, nerve root appearance, and conus level. Lumbar anatomical regions (annular fissure, disc desiccation, shortened pedicles, disc osteophyte, facet joint compression, and thecal sac diameter) were graded as normal, mild (nerve root is contacted on one side or surrounding fat effaced less than 90 degrees), moderate (nerve root contacted on two sides or fat effaced 90+ degrees), or severe (nerve root contacted on two sides, fat effaced 90+degrees, and nerve root visibly compressed). The clinician performing sensory and LE measurements was also blinded to MRI image results during phase 2 of testing.

## Statistical analyses

Data analyses were conducted using SPSS version 28.0(IBM Corp, Armonk, NY). A sample size of 40 subjects was estimated using a medium effect size (partial $\eta 2 = 0.06$), a power of 0.95, level of significance set at 0.05, and a 10% dropout rate. Data was summarized using frequencies and percentages for categorical variables, mean ±standard deviation (SD) for quantitative variables, and median (minimum, maximum) for ordinal variables. The normality of the quantitative outcomes was examined using Shapiro wilk test and normality plots. The subjects' baseline characteristics and outcome variables were compared between the two groups (LBP versus no LBP) using chi-square test of independence for qualitative variables, independent t-test for continuous variables, and Mann-Whitney U test for variables that were not symmetrical or ordinal. The outcome variables were compared by the presence of LBP (yes/no) and impingement (yes/no) using 2-way factorial analysis of variance (ANOVA). The mean outcomes for the 4 groups (LBP and impingement, LBP and no impingement, no LBP and impingement, and no LBP and no impingement) were compared using one way ANOVA. If results were significant then multiple comparisons were conducted using Bonferroni's adjustment. As there was a greater number of healthy males and a greater number of females with LBP, we controlled for the effect of gender using analysis of covariance. The level of significance was set at $p \leq 0.05$.

## Results

In this study, we aimed for 40 subjects, but enrolled a total of 32 subjects. Anticipated reasons for dropout included scheduling conflicts, other commitments, and lack of tolerance in a closed MRI tube. Of the 32 subjects recruited, 15 had chronic non-specific LBP, and 17 were healthy, asymptomatic controls. The subjects were recruited from local clinics, university, and community centers, and were screened to ensure they met the inclusion criteria. Of the 15 subjects who reported non-specific LBP, 12 subjects had local pain and 3 subjects had symptoms

(radiating pain) in their legs. The mean ± SD age of the subjects was 29.5±5.4 years, with a range of 21 to 45 years. The majority were males (n = 21; 65.6%) with a sex distribution of 65.6% male and 34.3% female. The mean body mass index (BMI) of subjects was 24.9±3.3kg/m$^2$. The distribution of subjects' characteristics by LBP is displayed in Table 1. The subject demographics were well-balanced between the two groups, with no significant differences observed in age, BMI, and co-morbidities (p>0.05). However, there was a significant association between sex and LBP (p = 0.034).

## Hamstring muscle length and straight leg raise comparison by LBP and impingement

There were no significant differences in mean hamstring length (HL) angulation and mSLR by LBP (yes/no) (p>0.05; Table 2). Additionally, there was no significant difference in mean HL by impingement (38.3±15.6 versus 44.8±9.4, p = 0.08, Cohen's d = 0.50). However, there was a significant difference in mean mSLR by impingement (yes/no) (57.6.3±8.7 versus 63.8±11.6, p = 0.05, Cohen's d = 0.60; Table 2). When comparing HL and mSLR among the four groups (no impingement and LBP, impingement and LBP, impingement and no LBP, no impingement, and no LBP), there was no significant difference among the means in those groups (p>0.05; Table 3).

## Sensory tests: Pain pressure threshold, vibration disappearance threshold, monofilament sensitivity

No change was detected in any subjects for mechanical pain threshold using pinprick stimulus and mechanical detection threshold assessment using Von-Frey filaments. These variables were excluded from statistical analysis.

When compared by the presence/absence of LBP, there were significant differences in mean ± SD PPT measured in the quadriceps left (QL), quadriceps right (QR) by LBP (14.2±4.0 versus 17.3±4.5, p = 0.022, Cohen's d = 0.73; and 15.1±5.3 versus 17.8±3.7, p = 0.05, Cohen's d = 0.60; respectively, Table 2). In addition, there were significant differences in the mean PPT measured in the tibialis anterior left (ATL) and medial calf left (MCL; (19.6±5.9 versus 22.4 ±2.8, p = 0.049, Cohen's d = 0.61; and 13.8±4.4 versus17.7±3.8, p = 0.006, Cohen's d = 0.95; respectively, Table 2) when compared by the presence/absence of LBP. However, there were no significant differences in mean PPT measured in the paraspinal left (PL), paraspinal right (PR), tibialis anterior right (ATR), medial calf right (MCR), instep left (IL), instep right (IR) by whether they had LBP or not (p>0.05, Table 2). For vibration (seconds), there were no significant difference in mean medial epicondyle left (MEL) and medial epicondyle right (MER) by whether they had LBP or not (p>0.05, Table 2). Results did not change after controlling for sex using analysis of covariance.

When compared by the presence/absence of MRI visible impingement, there were significant differences in mean ± SD PPT for PL and PR by impingement (21.6±3.2 versus18.9±4.8, p = 0.039, Cohen's d = 0.63 and 21.8±3.0 versus 19.4±4.5, p = 0.041, Cohen's d = 0.63; respectively). However, there were no significant difference in mean PPT measure in the QL, QR, ATL, ATR, MCL, MCR, IL, and IR between subjects with versus without impingement and those without (p>0.05, Table 2). For vibration (seconds), there were no significant differences in mean MEL and MER by whether they had impingement or not (p>0.05, Table 2).

There was no significant interaction between LBP (yes/no) and impingement(yes/no) for all vibration detection sights(p>0.05), except at the MER (p = 0.026, $\eta^2$ = 0.17). Bonferroni's post hoc comparisons showed that mean vibration measured at the MER of subjects with no

**Table 1. Baseline subjects' characteristics by group (N = 32).**

| Characteristic | No LBP ($n_1$ = 17) Frequency (%) | LBP ($n_2$ = 15) Frequency (%) | P-Value |
|---|---|---|---|
| Gender | | | |
| Female | 3 (17.6) | 8 (53.3) | 0.03 |
| Male | 14 (82.4) | 7 (46.7) | |
| Medication | | | |
| No | 16 (94.1) | 0 (0.0) | <0.001 |
| Yes | 1 (5.9) | 15 (100.0) | |
| Comorbidities | | | |
| Yes | 0 (0.0) | 4 (26.7) | 0.11 |
| No | 17 (100.0) | 11 (73.3) | |
| Rehabilitation | | | |
| No | 17 (100.0) | 10 (66.7) | 0.04 |
| Yes | 0 (0.0) | 5 (33.3) | |
| Age (years); mean ± SD | 29.4 ± 4.0 | 29.7 ± 6.7 | 0.85 |
| Onset* (years) | 0.0 (0,10) | 2.0 (0.25, 23) | <0.001 |
| VAS current pain* (0–10) | 0.0 (0, 3) | 2.0 (0, 5) | 0.01 |
| VAS worst pain *(0–10) | 0.0 (0, 8) | 5.0 (0. 8) | <0.001 |
| BMI (kg/m$^2$); mean ± SD | 24.6 ± 2.2 | 25.4 ± 4.2 | 0.52 |
| Occupation | | | |
| Student | 6 (35.3) | 8 (53.3) | 0.66 |
| Healthcare | 6 (35.3) | 6 (40.1) | |
| Education | 3 (17.7) | 1 (6.7) | |
| Administrative | 2 (11.7) | 0 (0.0) | |
| Education (by degree) | | | |
| Associates/ Bachelors | 10 (58.5) | 9 (60.0) | 0.99 |
| Doctorate | 4 (23.5) | 3 (20.0) | |
| Masters | 1 (5.9) | 1 (6.7) | |
| PhD | 2 (11.8) | 2 (13.3) | |
| Pain provoking triggers | | | |
| None | 16 (94.1) | 1 (6.7) | <0.001 |
| Stagnant positions | 1 (5.9) | 8 (53.3) | |
| Activity: Walking >30min, bending, lifting, pulling | 0 (0.0) | 7 (40.0) | |
| Outcome Measures | | | |
| PainDetect score (0–38) | | | |
| 0 to12: Unlikely neuropathic pain | 17 (100.0) | 14 (93.3) | <0.001 |
| 13 to 18: Possible neuropathic pain | 0(0.0) | 1 (6.7) | |
| 19 to 38: Likely neuropathic pain | 0 (0.0) | 0 (0.0) | |
| ODI ** | | | |
| None | 15 (88.2) | 0 (0.0) | <0.001 |
| Min | 2 (11.8) | 13 (86.7) | |
| Mod | 0 (0.0) | 1 (6.7) | |

**Abbreviation:** SD, standard deviation; BMI, body mass index; VAS, visual analogue scale from 0 to 10; ODI, Oswestry Disability Index.

*: median (minimum, maximum).

**: one subject in LBP group did not report ODI questionnaire.

**Table 2. Mean ± standard deviation of outcomes by LBP (Yes/No) and impingement (Yes/No).**

| Outcomes | LBP (n₁ = 15) | No LBP (n₂ = 17) | P-value (d) | Impingement (n₁ = 15) | No Impingement (n₂ = 17) | P-value (d) | P-value (LBP x Impingement) |
|---|---|---|---|---|---|---|---|
| Hamstring length (˚) | 45.6±10.9 | 38.4±13.9 | 0.06 (**0.6**) | 38.3±15.6 | 44.8±9.4 | 0.08 (**0.5**) | 0.31 (0.04) |
| Straight Leg Raise (˚) | 61.3±9.6 | 60.6±11.7 | 0.861 (0.06) | 57.6±8.7 | 63.8±11.6 | **0.05 (0.60)** | 0.533(0.01) |
| **Pain Pressure Threshold (kg/sec)** | | | | | | | |
| Paraspinal Left | 20.1±3.5 | 20.3±5.0 | 0.880 (0.05) | 21.6±3.2 | 18.9±4.8 | **0.039 (0.63)** | 0.172 (0.16) |
| Paraspinal Right | 20.0±3.5 | 21.0±4.5 | 0.510 (0.25) | 21.8±3.0 | 19.4±4.5 | **0.041 (0.63)** | 0.267 (0.13) |
| Quad Left | 14.2±4.0 | 17.3±4.5 | **0.022 (0.73)** | 16.6±4.5 | 15.2±4.6 | 0.376 (0.31) | 0.145 (0.17) |
| Quad Right | 15.1±5.3 | 17.8±3.7 | **0.050 (0.60)** | 17.9±4.8 | 15.4±4.3 | 0.063 (**0.55**) | 0.099 (0.20) |
| Tibialis Anterior Left | 19.6±5.9 | 22.4±2.8 | **0.049 (0.61)** | 22.2±4.2 | 20.1±4.9 | 0.100 (0.46) | 0.247 (0.14) |
| Tibialis Anterior Right | 19.8±3.9 | 21.3±4.2 | 0.307 (0.37) | 21.3±3.3 | 19.9±4.7 | 0.367 (0.34) | 0.232 (0.14) |
| Medial Calf Left | 13.8±4.4 | 17.7±3.8 | **0.006 (0.95)** | 16.9±4.3 | 14.9±4.5 | 0.215 (0.45) | 0.331(0.03) |
| Medial Calf Right | 14.8±3.4 | 16.8±4.9 | 0.093 (0.47) | 16.7±4.1 | 15.2±4.6 | 0.355 (0.34) | 0.527 (0.08) |
| Instep Left | 19.5±4.3 | 19.2±5.0 | 0.827 (0.06) | 20.3±3.6 | 18.5±5.4 | 0.294 (0.39) | 0.753 (0.04) |
| Instep Right | 19.0±4.5 | 20.4±4.6 | 0.400 (0.31) | 21.1±3.1 | 18.6±5.2 | 0.118 (**0.58**) | 0.419 (0.10) |
| **Vibration (seconds)** | | | | | | | |
| Medial Epicondyle Left | 7.7±4.0 | 7.2±4.0 | 0.725 (0.13) | 7.0±4.3 | 7.9±3.8 | 0.571 (0.22) | 0.253 (0.13) |
| Medial Epicondyle Right | 7.6±3.2 | 8.7±6.5 | 0.562 (0.21) | 6.7±3.5 | 9.6±6.0 | 0.059 (**0.60**) | **0.027 (0.16)** |
| Lateral Epicondyle Left | 6.4±2.9 | 6.9±4.8 | 0.716 (0.13) | 5.9±3.2 | 7.3±4.6 | 0.356 (0.25) | 0.103 (0.20) |
| Lateral Epicondyle Right | 7.0±3.4 | 6.6±3.8 | 0.787 (0.11) | 6.4±3.0 | 7.2±4.1 | 0.526 (0.22) | 0.511 (0.08) |
| Medial Malleolus Left | 9.2±4.9 | 7.0±4.6 | 0.104 (0.46) | 6.9±4.3 | 9.1±5.1 | 0.212 (0.47) | 0.338 (0.11) |
| Medial Malleolus Right | 9.0±5.1 | 7.0±4.0 | 0.110 (0.44) | 7.1±4.2 | 8.7±4.8 | 0.347 (0.35) | 0.365 (0.11) |
| Lateral Malleolus Left | 9.6±4.4 | 7.6±5.0 | 0.127 (0.43) | 7.8±4.5 | 9.2±5.0 | 0.419 (0.29) | 0.600 (0.06) |
| Lateral Malleolus Right | 8.8±3.4 | 7.0±4.0 | 0.092 (0.48) | 6.7±3.5 | 8.8±3.8 | 0.064 (**0.57**) | 0.227 (0.14) |

**Abbreviation:** LBP, low back pain; quad, quadricep.

impingement and no LBP significantly differed from those with no MR visible impingement and LBP (12.3±1.7 versus 7.2±1.6, p = 0.03).

When comparing outcomes among the four groups (no impingement and LBP, impingement and LBP, impingement and no LBP, no impingement, and no LBP), there was a significant difference in mean ± SD for PPT at the MCL among the four groups ($F_{3,28}$ = 3.1, p = 0.043). Bonferroni's post hoc comparisons showed that there was a significant difference in mean MCL between subjects with impingent and no LBP and those with no impingement and LBP (19.0±2.7 versus 13.8±4.7, p = 0.034). Also, there was a significant difference in mean vibration at the MER among the four groups ($F_{3,28}$ = 3.1, p = 0.045). Bonferroni's post hoc comparisons showed that there was a significant difference in mean between subjects with impingement and no LBP and those with no impingement and no LBP (5.6±3.2 versus 12.3 ±7.6, p = 0.020). However, there were no significant differences among the four groups for the remaining outcomes, mSLR and HL(p>0.05).

## Discussion

Neurodynamic tests such as the mSLR, and HL measurement, are clinical tests used to diagnose LBP. Specifically, a mSLR test can be used to diagnose pathoanatomical nerve root compression among other neural dysfunctions, and in this study, it was positively associated with lumbar nerve root impingement, but were not associated with LBP. Furthermore, a reduction in hamstring flexibility was seen in those with LBP, although the reduction was not statistically significant. With a moderate effect size suggesting practical significance, HL may be more

**Table 3. Mean ± standard deviation of outcomes by groups.**

| Outcomes | No impingement and LBP (n₁ = 9) | Impingement and LBP (n₂ = 6) | Impingement and no LBP (n₃ = 9) | No impingement and no LBP (n₄ = 8) | P-value ($\eta^2$) |
|---|---|---|---|---|---|
| Hamstring length (°) | 45.9±10.4 | 45.2±12.6 | 33.7±16.4 | 43.6±8.7 | 0.17 (0.16) |
| Straight Leg Raise (°) | 64.8±10.2 | 56.0±6.3 | 58.7±10.2 | 62.7±13.6 | 0.397 (0.10) |
| **Pain Pressure Threshold (kg/sec)** | | | | | |
| Paraspinal Left | 19.9±3.3 | 20.3±4.1 | 22.5±2.4 | 17.9±6.2 | 0.171 (0.16) |
| Paraspinal Right | 19.6±3.6 | 20.6±3.6 | 22.7±2.4 | 19.1±5.6 | 0.273 (0.13) |
| Quad Left | 13.0±3.5 | 15.8±4.4 | 17.1±4.7 | 17.6±4.7 | .145 (0.17) |
| Quad Right | 13.3±4.0 | 17.8±6.1 | 17.97±4.1 | 17.7±3.5 | .099 (0.20) |
| Tibialis Anterior Left | 18.5±5.8 | 21.3±6.1 | 22.9±2.4 | 21.8±3.3 | .247 (0.14) |
| Tibialis Anterior Right | 18.3±3.9 | 22.0±2.9 | 20.8±3.6 | 21.8±5.0 | .232 (0.14) |
| Medial Calf Left** | 13.8±4.7 | 13.8±4.4 | 19.0±2.7 | 16.2±4.4 | **.043** (0.25) |
| Medial Calf Right | 14.3±3.0 | 15.7±4.0 | 17.3±4.2 | 16.3±5.9 | .527 (0.08) |
| Instep Left | 18.7±4.6 | 20.8±4.0 | 20.0±3.5 | 18.3±6.5 | .753 (0.04) |
| Instep Right | 18.2±4.9 | 20.3±3.7 | 21.6±2.8 | 19.0±5.9 | .419 (0.05) |
| **Vibration (seconds)** | | | | | |
| Medial Epicondyle Left | 6.9±2.5 | 9.1±5.6 | 5.7±2.7 | 9.0±4.7 | .253(0.13) |
| Medial Epicondyle Right* | 7.2±3.0 | 8.4±3.6 | 5.6±3.2 | 12.3±7.6 | **.045**(0.25) |
| Lateral Epicondyle Left | 5.6±2.5 | 7.6±3.3 | 4.9±2.7 | 9.2±5.6 | .103(0.20) |
| Lateral Epicondyle Right | 6.5±3.8 | 7.7±2.9 | 5.5±2.9 | 8.0±4.5 | .511(0.08) |
| Medial Malleolus Left | 9.4±5.1 | 9.0±5.1 | 5.6±3.4 | 8.7±5.4 | .338(0.11) |
| Medial Malleolus Right | 8.8±5.5 | 9.3±4.8 | 5.7±3.3 | 8.5±4.3 | .365(0.11) |
| Lateral Malleolus Left | 9.8±5.0 | 9.4±3.6 | 6.8±5.0 | 8.6±5.2 | .600(0.06) |
| Lateral Malleolus Right | 9.0±3.7 | 8.4±3.2 | 5.6±3.4 | 8.5±4.3 | .227(0.14) |

**Abbreviation:** LBP, low back pain; quad, quadricep.

* Significant difference between impingement and no LBP versus impingement and no LBP.

** Significant difference between impingement and no LBP versus no impingement and LBP.

influential to LBP compared to those with or at risk for developing lumbar impingement. Individually these factors (HL, mSLR, lumbar nerve impingement) may relate to the LBP experience, thus investigating the combined association seems plausible. The interaction between mSLR, LBP, and lumbar nerve root impingement combined was compared in four study groups (no impingement and LBP, impingement and LBP, impingement and no LBP, no impingement, and no LBP) but no significant differences were detected. Although these results can provide valuable information, their interpretation should be made with caution, taking into account the study's design and limitations related to the number of participants in each group.

In this study the majority (80%) of subjects who reported having non-specific LBP had local symptoms only, meaning they did not experience any radiating pain in their legs. Three subjects, however, did report having LBP with radiating pain, which suggests that they may be experiencing some form of nerve compression or irritation that is causing the pain to radiate down their legs. Non-specific LBP without leg pain can be caused by a variety of factors, including nociceptive and neuropathic factors. Nociceptive pain is caused by tissue damage or inflammation, such as a muscle strain or sprain, and can be felt as a dull, achy pain. Neuropathic pain, on the other hand, is caused by damage or dysfunction to the nerves themselves.

Common symptoms of neuropathic pain include burning, tingling, sensitivity to touch and pressure, numbness, paresthesia, weakness, and loss of coordination [55]. When it comes to non-specific LBP without leg pain, it is often difficult to determine whether the pain is primarily nociceptive or neuropathic in nature. However, some common causes of non-specific LBP include muscle strain or sprain, degenerative disc disease, facet joint dysfunction, and myofascial pain syndrome.

Hamstring tightness is often observed with LBP, however all groups in our study exhibited significantly reduced hamstring extensibility. HL extensibility may alter lumbopelvic movement and subsequently lead to abnormal stress to the lumbar spine and supporting structures [56]. Using the UPA to assess HL flexibility, [57] found that the mean unilateral popliteal angle was 35.2 degrees (35-degrees short of full knee extension), with a standard deviation of 9.1 degrees. A previous study that used UPA found that angles greater than 45 degrees is indicative of hamstring stiffness [58]. Also, a prior study reported a minimal detectable change (MDC) of 3.7° for the popliteal angle in healthy young adults. The same study reported high intra-rater and inter-rater reliability for popliteal angle measurements [59]. These findings are clinically important and suggest that HL is reliably measured by UPA and limitations in HL flexibility may be a factor of LBP and/or an adaptation of pain. Our study found that individuals with LBP had the most limited hamstring flexibility with a medium effect size (Cohen's d of 0.6). This suggests that a difference of 7 degrees in HL is significant from a clinical perspective. A prior study demonstrated a positive association between HL flexibility and the severity of LBP [60]. Differing from the results of [59] study, all groups in our study had significant HL stiffness, low disability, and severity levels between the groups. Related to occupation, our study more than half of those with LBP were students which may factor into HL flexibility deficits. Other factors that could impact HL may involve a sedentary lifestyle, exercise habits, and opposing muscle group integrity. While these aspects were not explored in this study, they are important to consider when clinically assessing an individual with LBP.

Contrary to findings from prior research, our subjects with lumbar nerve root impingement demonstrated more hamstring flexibility compared to the no impingement group. [61] evaluated the effect of L5 nerve root irritation on hamstring flexibility in patients with radiculopathy. They reported that the L5 nerve root irritation significantly reduced hamstring flexibility compared to healthy controls, indicating that neurological involvement can affect hamstring flexibility. [62] showed that the presence of LBP, which can be associated with nerve root impingement, significantly affected the results of the popliteal angle test, indicating that neurological involvement can impact hamstring flexibility and range of motion. The difference in our study may be related to a combination of low pain levels, minimal disability scores, and the lack or leg symptoms in most subjects. HL may impact lumbar spine and pelvic mechanics in various ways making it difficult to specify causation of lumbar impingement. A study investigating hamstring flexibility and lumbopelvic mechanics found a positive correlation between reduced hamstring muscle flexibility and increased stress to posterior lumbar structures [63]. In our study, we did not differentiate between specific lumbar nerve root compressive dysfunctions, and it is possible subjects with impingement had alternative causes of impingement beyond hamstring length. Hamstring muscles contribute to lumbopelvic stability [55, 64], and our findings of increased hamstring flexibility could be attributed to ineffective pelvic stability influenced by hamstring length. Taken together, the reduction of knee extension during the UPA could be related LBP but may not be a clinically relevant finding for lumbar impingement without radiating symptoms.

The hypothesis that those with chronic LBP, with or without radiating pain, would have a positive correlation with findings of the mSLR test was not supported by the present study; there was no significant relationship observed between chronic LBP and the neurodynamic

test when compared to healthy subjects. Chronic LBP without leg symptoms is often considered nociceptive due to the location of the symptoms. However, several studies suggest that chronic musculoskeletal disorders may have a neural component contributing to pain persistence, which may not be limited to traditional radiculopathy patterns [26, 65–67]. In our study only 20% (3/15) of our LBP subjects had leg pain, yet 46.6% (7/15) of subjects had a reduced SLR angle below the mean for those with LBP. The common practice of implementing a mSLR test exclusively for individuals experiencing leg symptoms may need to be reevaluated, as nearly half of the subjects with LBP without leg symptoms had a decreased mSLR angle.

The association between the mSLR and lumbar nerve root impingement can be explained with the fundamental understanding that the mSLR induces neural tension. In the present study, we found that subjects with lumbar nerve root impingement had significantly lower mSLR angulation (57.6±8.7 degrees) compared to those without impingement (63.8±11.6 degrees). [68] found a mean mSLR angle of 48.5–48.9 degrees in asymptomatic subjects with a mean age of 36.9 years old. Compared to this study, the lumbar nerve impingement group also included asymptomatic individuals, however the mSLR angulation difference may be a result of a lowered mean age (29.7 years old). Another study found a mean of 50 degrees in those with LBP with or without leg symptoms but ability to reproduce symptoms [33]. In the same study, those that were not able to reproduce symptoms had a mean SLR angle of 60–62 degrees in those with or without leg symptoms. Although 93% of subjects with impingement did not have radiating symptoms, the decreased angulation may be explained by neural tissue mechanosensitivity.

Within this study, a protocol to determine an appropriate time to measure the mSLR angulation was developed rather than assigning a positive or negative result. Systematic reviews [69–72] on the diagnostic accuracy of clinical neurological test in those with lumbar radiculopathy, disc herniation, and/or sciatica suggested that the diagnostic utility of neurodynamic tests were inconsistent and still debatable amongst researchers. Discrepancies in performance of the neurodynamic tests and defining positive results were reasons that lead to the proposed varied diagnostic ability. [73] investigated whether a mSLR, performed with the addition of ankle dorsiflexion, was associated with lumbar pathology seen on MRI. The results demonstrated strong association with pathology seen on MRI and high validity in detecting neural symptoms. Using the mSLR to reproduce symptoms, 85% of their subjects had either lumbar disc herniation or lumbar nerve root compression of L4-5 or L5-S1. Similar to the involved lumbar segments reported by [73], the most common region for lumbar nerve root impingement in our study was L4-5, L5-S1, and L3-4.

Recognized in the Clinical Practice Guidelines for LBP management from the Academy of Orthopedic Physical Therapy and the American Sports Physical therapy of the American Physical Therapy Association (APTA), MR images have frequent false positive/negative results which limits effectiveness in identifying anatomic pain generators [15]. The lack of correlation between image and symptoms was demonstrated in the present study. Lumbar nerve root impingement was seen in both the LBP and healthy control group with 52.9% (9/17) of those with lumbar nerve root impingement being asymptomatic. This incongruency highlights the need for alternative diagnostic tests to specify pain mechanism. Among subjects who were pain-free with MRI findings of nerve encroachment, 66.6% (6/9) or two-thirds, had a reduced mSLR angulation below the mean for that group.

A systematic review included 33 articles that reported imaging findings for 3110 asymptomatic individuals [16]. Their findings revealed that the prevalence of disc degeneration in asymptomatic individuals increased with age, with 37% of 20-year-olds having disk degeneration compared to 96% of 80-year-olds. Similarly, the prevalence of disc bulge increased from 30% to 84%, while the prevalence of disc protrusion increased from 29% to 43% in individuals

aged 20 to 80 years. [16] also found that the prevalence of annular fissure increased from 19% to 29% in individuals aged 20 to 80 years. Their results indicate that age is a significant factor in the prevalence of degenerative spinal changes in asymptomatic individuals. Our findings are supported by [16] with 28% (9/32) of our total subjects having MRI defined lumbar nerve root impingement and asymptomatic. Corresponding to this study, it did not come as a surprise to have asymptomatic subjects with MRI defined nerve root impingement. While changes in lumbar spine are common on MRI it may not predict neuropathic pain or pain intensity [74, 75]. Considering the mixed utility of an MRI, and the findings of the present study related to a reduced mSLR test angulation associated with lumbar nerve root impingement, an explanation could be that the mSLR test is a barometer for nerve root inflammation, and a probable clinical sign for neural component contributing to the pain experience.

Sensation tests are commonly performed as diagnostic aids to assess neuropathic components to the pain experience and assist with a pathoanatomical diagnosis formulation. In our study, we found an association between LBP and a decrease in PPT in bilateral quadricep muscles, ATL, and the MCL. PPT testing in the impingement group was not significant in sites distal to the lumbar spine suggesting that lumbar impingement findings alone cannot account for distal symptoms. Prior studies investigating chronic LBP and PPT have found a similar decrease in PPT tolerance [75–77]. Despite a significant difference between the LBP group and a similar decrease in [77], the difference in our study was small. Comparing studies, this could be due to lower tissue irritability levels and a younger mean age in the chronic LBP group. A study investigating sensitivity of lower leg regions and nerve trunk areas used the PPT test in a unilateral chronic heel pain population which demonstrated bilateral widespread pressure sensitivity in both lower extremity nerve trunk and musculoskeletal structures [78].The widespread bilateral lower limb sensitivity observed by [77] was suggested to be due to centrally mediated nociceptive pain processing, based on pain location, PPT test, and correlation to foot function disability index. Altered pain processing for nociception cannot be inferred in our study, as we did not test a significantly distal site away from the lower extremity [79]. Also, in-depth musculoskeletal assessment was not performed to be able to rule out alternative causes for lowered PPT findings.

Other somatosensory tests such as mechanical detection threshold by using monofilaments, mechanical pain thresholds by pinprick, and vibration disappearance thresholds were assessed and no significant difference were found between the groups, most likely related low pain intensity and minimally reported disability (ODI). Although no differences were found, related to sensation, [37] reported that monofilaments had good test-retest reliability and inter-rater reliability when used to assess mechanical detection thresholds in healthy subjects. The validity of filaments has also been supported by studies that have shown a strong correlation between filament measurements and other measures of sensory function, such as QST and nerve conduction studies [80–83]. Four lower extremity sites were used for vibration detection threshold assessment and the results showed a reduced threshold trend in only the impingement group, however differences were not significant. Polyneuropathy was defined with abnormal vibration sense in 2 or 4 regions with a specificity of 94% and sensitivity of 63% [54]. According to [48], the threshold for vibration detection in healthy children and young adults is 15 seconds, with this threshold declining with age even in the absence of peripheral neuropathy. All subjects were below this threshold likely related to age.

Pain processing in general is complex and can become a vicious cycle resulting in poor inhibition of pain impulses; a process that cannot be determined by MRI. Future studies analyzing brain activity using fMRI and nociceptive pain are warranted to better understand the pain experience in the chronic LBP population. Additionally, to build upon the findings of the

current study, utilizing fMRI with the mSLR test may highlight pain processing details within a specific population.

The current study has certain limitations that should be considered when interpreting the findings. One such limitation is that the study participants had chronic LBP of more than 3 months duration, but they reported low levels of pain and disability, indicating low tissue irritability/reactivity. It is possible that the findings of the study may not be generalizable to individuals with higher levels of tissue irritability or pain. Second, peripheral nerve entrapment along the course of the peripheral nerves may have been present and was not accounted for. Third, only 3 subjects had subjectively reported radiating leg pain therefore usage of the Pain-Detect for neuropathic pain diagnoses was not as useful. Fourth, the majority of our low back subjects likely did not have neuropathic pain as identified by the PainDetect questionnaire while only one subject had possible neuropathic pain. Lastly, larger sample size is needed to confirm trends found and assist with generalizability of the findings. The small sample size and the imbalance among subgroups affects the ability to generalize the findings. While a larger sample size is warranted, an inherent financial limitation is imposed with imaging studies and may influence sample size and extent of data collection. Despite all discussed challenges, this study provides valuable insights within the forementioned limitations.

Based on the findings of the present study, the mSLR test was found to be significantly associated with lumbar nerve root impingement, regardless of the presence of radiating leg symptoms. Clinically, these findings suggests that the mSLR test may be a useful neurodynamic test to assess nerve root impingement in patients with chronic nonspecific LBP with low tissue reactivity/irritability (LBP and low disability) who are unlikely to have neuropathic pain as identified by self-report. The traditional use of a neurodynamic test based on symptom location is debatable based on the findings which proposes clinical application of the mSLR during the physical objective assessment of lumbar nerve root impingement irrespective of leg symptoms. Reduced hamstring extensibility, as measured by the UPA, was greatest in individuals with LBP and inversely related to those with impingement. With this observation, hamstring muscle length may not be a critical influencer in the development of impingement. In clinical practice assessing lumbar impingement, objective exam attention should be directed towards identifying alternate lumbopelvic compensatory patterns or factors aside from hamstring length. Additionally, the study found a relationship between chronic LBP and PPT, indicating a greater sensitivity to pain although the tissue source is not known. The findings of this study, however, highlights the diagnostic challenges that exist in the chronic nonspecific LBP population with uncorrelated neuroanatomical image findings and specific clinical tests. Therefore, the authors recommend that an open-minded and inclusive approach be maintained towards the multifactorial etiology underlying the manifestation of symptoms and their physical manifestations. Overall, this study contributes to the understanding of chronic nonspecific LBP symptoms and provides insights into the usage of current clinical tests.

The findings in this study may serve as a catalyst for future studies to understand the link between chronic LBP, neurologically mediated pain experience, and diagnostic clinical tests. Future studies should have a more inclusive age range to validate the findings in different age groups. Future research should consider including participants with a wider range of pain levels, tissue reactivity and radiating/neuropathic pain to better understand the relationship between mSLR angulation, hamstring muscle length angulation, and LBP. In addition, future research may focus on investigating the pathoneurodynamics that underlie these associations. Pathoneurodynamics refers to the interplay between neurological, mechanical, and pathological factors that contribute to the development and progression of peripheral neuropathic disorders. By examining the pathoneurodynamics of lumbar nerve root impingement and its relationship to mSLR angulation and hamstring muscle length, researchers may gain a better

understanding of the underlying mechanisms of the condition. Expanding on the current study, distinguishing between specific anatomical causes of lumbar nerve root compression and comparing them to hamstring length and the neurodynamic test will enhance our understanding of these factors in relation to LBP. Further development based on this study may pave the way for more effective diagnostic and treatment strategies for those with LBP and lumbar nerve root impingement. Also, other factors such as gender and physical activity levels may also influence the relationship between these variables and should be considered in future research.

## Conclusion

The purpose of this study was to compare mean mSLR and hamstring muscle length (HL) between chronic non-specific LBP and healthy subjects and secondly to compare mean mSLR between those with lumbar impingement and those without as determine by MRI. Lumbar nerve impingement was associated with a mSLR test regardless of the presence of leg symptoms. Hamstring flexibility is related to non-specific LBP and may not be an important clinical finding in diagnosis lumbar nerve root impingement. It may be necessary to reconsider the common practice of implementing a mSLR test exclusively for individuals experiencing leg symptoms. The findings add to a deepened understanding for the usage of a mSLR in those with chronic non-specific LBP.

## Acknowledgments

We would like to thank the Loma Linda University Medical Center radiology department for supporting the investigators as well as the participants contributing their time to advance scientific research.

## Author Contributions

**Conceptualization:** Joan Hermosura, Brenda Bartnik-Olson, Jonathan Venezia.

**Data curation:** Joan Hermosura, Brenda Bartnik-Olson, Noha Daher.

**Formal analysis:** Joan Hermosura, Noha Daher.

**Funding acquisition:** Everett Lohman, III.

**Investigation:** Joan Hermosura, Everett Lohman, III, Brenda Bartnik-Olson, Jonathan Venezia.

**Methodology:** Joan Hermosura, Everett Lohman, III, Brenda Bartnik-Olson.

**Project administration:** Joan Hermosura, Everett Lohman, III, Brenda Bartnik-Olson.

**Resources:** Joan Hermosura, Everett Lohman, III, Brenda Bartnik-Olson.

**Supervision:** Joan Hermosura, Everett Lohman, III, Brenda Bartnik-Olson.

**Validation:** Noha Daher.

**Writing – original draft:** Joan Hermosura, Everett Lohman, III.

**Writing – review & editing:** Joan Hermosura, Everett Lohman, III, Brenda Bartnik-Olson, Jonathan Venezia, Noha Daher.

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
