## [Decision Letter · Decision Letter 0]

20 Nov 2023

PONE-D-23-29503The usage of a modified straight-leg raise neurodynamic test and hamstring flexibility to aid in the diagnosis of non-specific low back painPLOS ONE

Dear Dr. Hermosura,

Thank you for submitting your manuscript to PLOS ONE. After careful consideration, we feel that it has merit but does not fully meet PLOS ONE’s publication criteria as it currently stands. Therefore, we invite you to submit a revised version of the manuscript that addresses the points raised during the review process.

We look forward to receiving your revised manuscript.

Kind regards,

Ravi Shankar Yerragonda Reddy, Ph.D

Academic Editor

PLOS ONE

Journal Requirements:

"EL

12,000

Dr. Craig Jackson

https://researchaffairs.llu.edu/

NO"

6. Please amend the manuscript submission data (via Edit Submission) to include authors Dr. Brenda Bartnik-Olsen, Dr. Noha Daher, and Dr.Jonathan Venezia.

Additional Editor Comments:

Reviewer's Report:

Abstract:

1. The abstract provides a clear overview of the study's objectives, methods, results, and conclusions. However, it lacks a brief statement of the study's significance or implications for clinical practice.

Introduction:

1. The introduction provides extensive background information on chronic low back pain (LBP), its prevalence, and the challenges in diagnosing and treating it. While this context is important, it feels somewhat overwhelming and could benefit from more concise and focused framing of the research problem.

2. The introduction mentions the International Association for the Study of Pain (IASP) classification system, but it doesn't explain how this system relates to the study or its objectives. Providing this connection would help readers understand why the IASP classification is relevant to the research.

3. There are several grammatical issues, such as missing commas, that need to be addressed to improve the overall readability of the introduction.

Methods:

1. The methods section provides a comprehensive overview of the study design, inclusion/exclusion criteria, and data collection procedures. However, there are some issues:

2. The section on participant recruitment mentions that recruitment began in September 2020 and terminated in July 2022. This raises questions about the timeline of the study. Was data collection ongoing for nearly two years? Clarification is needed.

3. In the inclusion criteria for the Back Pain Group, it mentions "history of LBP symptoms greater than 3 months," but it's not clear how this duration was determined or verified. Providing details on how the chronicity of LBP was assessed would be helpful.

4. While the inclusion and exclusion criteria are mentioned, there's no information on the actual number of participants recruited or the final sample size. Reporting this information is essential.

5. The section describing the somatosensory and range of motion assessments is quite detailed, which is good for transparency. However, it might be helpful to provide a brief rationale for each assessment to explain why these specific measures were chosen.

6. The use of abbreviations like ODI, PD-Q, and QST is introduced in the Methods section but should be defined upon first use for clarity.

7. The details about the MRI imaging procedure are extensive, but it's not clear how the MRI results were analyzed or interpreted. Providing information on the criteria used to determine "impingement" or "no impingement" based on MRI findings would enhance understanding.

8. In the statistical analyses section, the mention of a 40% dropout should be explained. Why was such a high dropout rate anticipated? Was this based on previous studies or clinical experience?

Results and discussion

1. Sample Size and Group Imbalance: The study recruited a total of 32 subjects, with 15 subjects having chronic non-specific LBP and 17 healthy controls. The sample size appears relatively small, particularly when subgrouping for impingement and LBP. Additionally, the distribution of subjects across these subgroups is imbalanced, which could affect the statistical power and generalizability of the findings. This imbalance should be addressed or at least discussed in more detail to assess its potential impact on the results.

2. Lack of Clear Clinical Implications: While the study explores various factors related to LBP, such as hamstring flexibility, lumbar nerve root impingement, and neurodynamic tests, the manuscript does not clearly discuss the clinical implications of these findings. It is essential to bridge the gap between research and clinical practice by explaining how the results might impact the diagnosis or management of LBP in real-world scenarios.

3. Inconsistent Findings: The Results section presents some inconsistent findings, such as the lack of a significant difference in mean hamstring length by impingement and the significant difference in mean mSLR by impingement. These inconsistencies need to be thoroughly discussed and explained in the Discussion section. It's essential to consider potential confounding factors or limitations that could have contributed to these discrepancies.

4. Limited Discussion of Existing Literature: The Discussion section could benefit from a more extensive review of the existing literature on neurodynamic tests, hamstring flexibility, and their relationship with LBP. Providing a broader context by discussing previous studies and their findings would help readers understand where this study fits into the existing body of knowledge.

5. Lack of Discussion on Clinical Relevance: The manuscript should include a discussion on the clinical relevance of the findings. How do the results of this study inform clinical practice? Are there potential implications for diagnosing or managing LBP in patients? This aspect is crucial for translating research findings into practical applications.

6. Need for Future Directions: The Discussion section should conclude with a clear identification of potential avenues for future research. What questions remain unanswered, and how can future studies build upon the current findings to advance our understanding of LBP and related factors?

Reviewers' comments:

Reviewer's Responses to Questions

**Comments to the Author**

1. Is the manuscript technically sound, and do the data support the conclusions?

Reviewer #1: Yes

Reviewer #2: Yes

Reviewer #3: Yes

2. Has the statistical analysis been performed appropriately and rigorously? 

Reviewer #1: Yes

Reviewer #2: No

Reviewer #3: Yes

3. Have the authors made all data underlying the findings in their manuscript fully available?

Reviewer #1: Yes

Reviewer #2: Yes

Reviewer #3: Yes

4. Is the manuscript presented in an intelligible fashion and written in standard English?

Reviewer #1: Yes

Reviewer #2: Yes

Reviewer #3: Yes

5. Review Comments to the Author

Reviewer #1: The authors conducted significant study to investigate usage of a modified straight-leg raise neurodynamic test and hamstring flexibility to aid in the diagnosis of non-specific low back pain. The manuscript is well-written, and the methodology seems good. However, some comments should be answered before considering this manuscript for publication in PLOS ONE.

1. “Furthermore, hamstring flexibility was also associated with lumbar nerve root compression” line 396-397. This statement contradicts the finding in the result section (Table 2) that showed no significant differences by LBP or impingement. Please clarify.

2. “In this study the majority (80%) of subjects reported having non-specific LBP only,

meaning they did not experience any radiating pain in their legs. Three subjects, however, did report having LBP with radiating pain, which suggests that they may be experiencing some form of nerve compression or irritation that is causing the pain to radiate down their legs” line 406- 409. In our study only 20% (3/15) of our LBP subjects had leg pain,” line 453-454. These details are not reported in the results section, particularly under the characteristics of the subjects. Suggestion that this information be included in the results section.

3. Justify all paragraphs in this manuscript to ensure that information is well organized

Reviewer #2: Review comments on Manuscript Number: PONE-D-23-29503. “The usage of a modified straight-leg raise neurodynamic test and hamstring flexibility to aid in the diagnosis of non-specific low back pain"

Overall, the idea of research is very interesting to be studied nowadays and paper is coherently developed. However, there are some comments and suggestions.

General comments the authors need to review the manuscript grammar and editing.

Title

- You may consider change the title to [The usage of a modified straight-leg raise neurodynamic test and hamstring flexibility for diagnosis of non-specific low back pain]

Abstract

- Don’t use abbreviations in abstract unless you first write the full terms. Correct [mSLR, LBP, HL]

- The methods section should include information about the study sample [number of participants and age]

- It is recommended to write 5- 7 keywords in alphabetical order.

Introduction

- Well-structured but needs to be more concise.

Subjects and methods

- The age range is relatively wide from 18 to 50. I am afraid that many factors may influence your results

Statistical analysis

- Well structured

Discussion

- Well structured

Reviewer #3: The title of the manuscript should contain the study type. Conclusion of the abstract contained should be re-written to explain the results more obviously. The abstract should not contain apprevations. The study objectives should be declared obviously at the end of the introduction. The author didn't mention number of participants. Age range of 18 -45 as inclusion criteria is very wide range.

6. PLOS authors have the option to publish the peer review history of their article (what does this mean?). If published, this will include your full peer review and any attached files.

Reviewer #1: **Yes: **Rabiatul Adawiah Abdul Rahman

Reviewer #2: No

Reviewer #3: No

---

## [Author Response · Author response to Decision Letter 0]

28 Dec 2023

Reviewer 1:

1. “Furthermore, hamstring flexibility was also associated with lumbar nerve root compression” line 396-397. This statement contradicts the finding in the result section (Table 2) that showed no significant differences by LBP or impingement. Please clarify.

Thanks for your comment. I revised and expanded statement to “Furthermore, a reduction in hamstring flexibility was seen in those with LBP, although the reduction was not statistically significant. With a moderate effect size suggesting practical significance, HL may be more influential to LBP compared to those with or at risk for developing lumbar impingement” (track change lines 586-589). Hamstring flexibility was reduced in those with LBP with no statistical significance. However, effect size was moderate (d-0.6) suggesting clinical/practical significance.

2. “In this study the majority (80%) of subjects reported having non-specific LBP only,

meaning they did not experience any radiating pain in their legs. Three subjects, however, did report having LBP with radiating pain, which suggests that they may be experiencing some form of nerve compression or irritation that is causing the pain to radiate down their legs” line 406- 409. In our study only 20% (3/15) of our LBP subjects had leg pain,” line 453-454. These details are not reported in the results section, particularly under the characteristics of the subjects. Suggestion that this information be included in the results section.

Thank you. I added this information in the results section and expanded on the details as recommended (track change lines 485-497).

Reviewer 2: 

3. You may consider change the title to [The usage of a modified straight-leg raise neurodynamic test and hamstring flexibility for diagnosis of non-specific low back pain]

a. The title was revised to “The usage of a modified straight-leg raise neurodynamic test and hamstring flexibility for diagnosis of non-specific low back pain: A cross-sectional study.”

4. Abstract :Don’t use abbreviations in abstract unless you first write the full terms. Correct [mSLR, LBP, HL]

a. Thank you for your valuable feedback. I wrote full terms for mSLR, HL, and LBP in the abstract prior to usage. 

5. Abstract: The methods section should include information about the study sample [number of participants and age]

a. I added the number of subjects and age range detail to the methods section in the abstract (track change lines 39-40). 

6. Abstract: it is recommended to write 5- 7 keywords in alphabetical order. 

a. Thanks. I put keywords in alphabetical order (track change lines 64-65).

7. Introduction: Well-structured but needs to be more concise. 

a. Thank you for your valuable input. I revised introduction for increased conciseness. Revision included deleting unnecessary statements that did not support study objectives.

8. Subjects and methods: The age range is relatively wide from 18 to 50. I am afraid that many factors may influence your results

a. Thank you. We chose this age range because arthritic prevalence is less common among subjects less than 50 years of age. The mean age of subjects in this study was 29.5 ± 5.4. I have added a statement in for future studies with the recommendation for more inclusive age ranges (lines 866-867).

Reviewer 3:

9. The title of the manuscript should contain the study type.

a. Thanks. I have included the study design in the title.

10. Conclusion of the abstract contained should be re-written to explain the results more obviously. 

a. Thank you. I have clarified the main finding in the abstract conclusion (track change lines 50-51).

11. The abstract should not contain abbreviations.

a. Thanks. Abbreviations were removed from abstract, or first stated and then preceded with abbreviation (track change lines 32-33).

12. The study objectives should be declared obviously at the end of the introduction

a. Thanks. The objectives are reported and re-stated for clarity in the last paragraph of the introduction (track change lines 196-200).

13. The author didn't mention number of participants. 

a. Thank you. Number of participants were added to the abstract (track change lines 39-40).

Additional Editor comments:

14. The abstract provides a clear overview of the study's objectives, methods, results, and conclusions. However, it lacks a brief statement of the study's significance or implications for clinical practice.

a. Thank you. I added clinical implication to abstract (lines 59-61).

15. Introduction: The introduction provides extensive background information on chronic low back pain (LBP), its prevalence, and the challenges in diagnosing and treating it. While this context is important, it feels somewhat overwhelming and could benefit from more concise and focused framing of the research problem. 

a. Thank you for your feedback. The introduction was modified to assist with conciseness and flow.

16. Introduction: The introduction mentions the International Association for the Study of Pain (IASP) classification system, but it doesn't explain how this system relates to the study or its objectives. Providing this connection would help readers understand why the IASP classification is relevant to the research

a. Thank you. The IASP classification statements have been revised for conciseness and improved clarity in order to better support reasons for objectives (track change lines 128-130).

17. Introduction: There are several grammatical issues, such as missing commas, that need to be addressed to improve the overall readability of the introduction.

a. Thank you. Introduction has been reviewed. 

18. Methods: The section on participant recruitment mentions that recruitment began in September 2020 and terminated in July 2022. This raises questions about the timeline of the study. Was data collection ongoing for nearly two years? Clarification is needed.

a. Thank you for highlighting. Due to the global pandemic of COVID-19, research collection that included close participant proximity at Loma Linda University was paused temporarily leading to a lengthened recruitment/data collection period. Added statement to explain lengthened time of study (track change lines 262-264).

19. Methods :In the inclusion criteria for the Back Pain Group, it mentions "history of LBP symptoms greater than 3 months," but it's not clear how this duration was determined or verified. Providing details on how the chronicity of LBP was assessed would be helpful.

a. Thank you. Inclusion for the LBP group was based on the common definition of “chronic” which includes pain > 3months. Pain was self-reported and taken based on subjects’ assumed honest integrity. Those who experienced chronic symptoms and met other inclusion criteria were placed in the LBP group. Track change line 265-266 and 270-271 was added for clarity :“Back pain (or lack thereof) was solely self-reported and accepted at face value” and “Participants experiencing back pain consistently for >3months were considered to have chronic symptoms.”

20. While the inclusion and exclusion criteria are mentioned, there's no information on the actual number of participants recruited or the final sample size. Reporting this information is essential. 

a. Thank you. Total number of subjects were added to the methods section (track change line 265-266). This information is also found in the more detail in the results section (track change lines 485 -488).

21. The section describing the somatosensory and range of motion assessments is quite detailed, which is good for transparency. However, it might be helpful to provide a brief rationale for each assessment to explain why these specific measures were chosen.

a. The battery of somatosensory tests was based on Qualitative Sensory Test (QST). Site specific locations were based on lumbar spine neurological sensory levels (track change lines 300-309).

b. For Mechanical detection threshold assessment, we used monofilaments as they are commonly used for sensory testing in neurological and musculoskeletal disorders and are considered reliable and valid (track change lines 344-349).

c. Mechanical pain threshold assessment was also part of the QST battery to assess the integrity of fibers specific for sharp, acute, and protective pain (track change lines 357-360).

d. Pain pressure threshold assessment is also part of QST and assesses for deep pressure to pain tolerance. Added “Individuals with nerve compromise may have an altered tolerance to pressure.” (track change lines 371-374).

e. Vibration detection threshold assessment via tuning fork is cost effective and a commons sensory test in neuropathic conditions (track change lines 390-392).

f. We have added a statement regarding hamstring muscle length relation to mechanical forces. “Hamstring length may alter alignment and mechanics of the lumbopelvic region” (track change line 408).

g. mSLR was performed to maximally tension the lower extremity nerves to assess the mobility and sensitivity of neural tissue. (track change lines 413-416; 429-430). 

22. The use of abbreviations like ODI, PD-Q, and QST is introduced in the Methods section but should be defined upon first use for clarity.

a. Abbreviations for ODI and PD-Q were removed and postponed to the section that began to describe the questionnaires: ‘Phase 1: subjective demographics and questionaries.’ (track change lines 314). 

23. The details about the MRI imaging procedure are extensive, but it's not clear how the MRI results were analyzed or interpreted. Providing information on the criteria used to determine "impingement" or "no impingement" based on MRI findings would enhance understanding.

a. Thank you. Details on the criteria for impingement/no impingement as well as neuroradialogist qualification for reading images were added (track change lines 450-459).

24. In the statistical analyses section, the mention of a 40% dropout should be explained. Why was such a high dropout rate anticipated? Was this based on previous studies or clinical experience?

a. Thank you for your valuable input. This was a typing error. We estimated a 10% dropout rate based on clinical and research experience and nature of study. (track change lines469). The study included a closed MRI procedure and some participants were reluctant for personal reasons once in the imaging machine. Other reasons included scheduling conflict and other time commitments (track change lines 485-487).

25. Results and discussion: Sample Size and Group Imbalance: The study recruited a total of 32 subjects, with 15 subjects having chronic non-specific LBP and 17 healthy controls. The sample size appears relatively small, particularly when subgrouping for impingement and LBP. Additionally, the distribution of subjects across these subgroups is imbalanced, which could affect the statistical power and generalizability of the findings. This imbalance should be addressed or at least discussed in more detail to assess its potential impact on the results.

a. Thank you for your comment. Due to feasibility issues and the COVID-19 pandemic, it was hard to recruit participants. A statement discussing limitations to sample size and group imbalances were added ( track change lines 810-814).

26. Lack of Clear Clinical Implications and discussion on clinical relevance: While the study explores various factors related to LBP, such as hamstring flexibility, lumbar nerve root impingement, and neurodynamic tests, the manuscript does not clearly discuss the clinical implications of these findings. It is essential to bridge the gap between research and clinical practice by explaining how the results might impact the diagnosis or management of LBP in real-world scenarios. How do the results of this study inform clinical practice? Are there potential implications for diagnosing or managing LBP in patients? This aspect is crucial for translating research findings into practical applications.

a. Thank you. I have expanded the paragraph discussing applicability. Based on the findings, clinically the mSLR may be a useful test to assess those with nerve root impingement regardless of lower extremity symptoms and low symptom irritability. Hamstring extensibility was reduced in those with LBP and inversely related to lumbar impingement. Clinically, this suggests that attention to alternative factors aside from hamstring length may be more of a priority during the objective assessment of someone with lumbar impingement (track change lines 817-822; 853-855).

27. Inconsistent findings : The Results section presents some inconsistent findings, such as the lack of a significant difference in mean hamstring length by impingement and the significant difference in mean mSLR by impingement. These inconsistencies need to be thoroughly discussed and explained in the Discussion section. It's essential to consider potential confounding factors or limitations that could have contributed to these discrepancies.

a. Thank you. Clarified the finding on hamstring length and LBP in discussion. Hamstring length was most limited in those with LBP although not statistically significant (track change lines 586-589). Confounding factors for hamstring flexibility were acknowledged and added (track change lines 637-640). 

b. Factors that relate the lack of hamstring length significance to lumbar impingement may be related to a combination of factors including low pain levels, minimal disability score, and lack of leg symptoms (track change lines 654-656).

28. Limited Discussion of Existing Literature: The Discussion section could benefit from a more extensive review of the existing literature on neurodynamic tests, hamstring flexibility, and their relationship with LBP. Providing a broader context by discussing previous studies and their findings would help readers understand where this study fits into the existing body of knowledge.

a. Thank you. I added more context related to lumbopelvic mechanics, hamstring flexibility, and low back pain (track change lines 657-664).

29. Need for Future Directions: The Discussion section should conclude with a clear identification of potential avenues for future research. What questions remain unanswered, and how can future studies build upon the current findings to advance our understanding of LBP and related factors?

a. Thank you. I moved future research information to end of discussion and modified to include more specifics (track change lines 864-881).

---

## [Decision Letter · Decision Letter 1]

23 Jan 2024

The usage of a modified straight-leg raise neurodynamic test and hamstring flexibility for diagnosis of non-specific low back pain: A cross-sectional study

PONE-D-23-29503R1

Dear Dr. Joan Hermosura,

We’re pleased to inform you that your manuscript has been judged scientifically suitable for publication and will be formally accepted for publication once it meets all outstanding technical requirements.

Kind regards,

Ravi Shankar Yerragonda Reddy, Ph.D

Academic Editor

PLOS ONE

Additional Editor Comments (optional):

Based on the authors' thorough response to the reviewers' comments in the revised manuscript, I recommend accepting it for publication in its current form.

Reviewers' comments:

Reviewer's Responses to Questions

**Comments to the Author**

1. If the authors have adequately addressed your comments raised in a previous round of review and you feel that this manuscript is now acceptable for publication, you may indicate that here to bypass the “Comments to the Author” section, enter your conflict of interest statement in the “Confidential to Editor” section, and submit your "Accept" recommendation.

Reviewer #2: All comments have been addressed

2. Is the manuscript technically sound, and do the data support the conclusions?

Reviewer #2: Yes

3. Has the statistical analysis been performed appropriately and rigorously? 

Reviewer #2: Yes

4. Have the authors made all data underlying the findings in their manuscript fully available?

Reviewer #2: Yes

5. Is the manuscript presented in an intelligible fashion and written in standard English?

Reviewer #2: (No Response)

6. Review Comments to the Author

Reviewer #2: Review comments on Manuscript Number: (PONE-D-23-29503R1) entitled’The usage of a modified straight-leg raise neurodynamic test and hamstring flexibility for diagnosis of non-specific low back pain: A cross-sectional study’.

Overall, this study provides a novel approach. The idea of research is very interesting, well written and reasonable. I have no comments regarding the manuscript and recommend accepting it.

7. PLOS authors have the option to publish the peer review history of their article (what does this mean?). If published, this will include your full peer review and any attached files.

Reviewer #2: No

---

## [Editor Report · Acceptance letter]

20 Mar 2024

PONE-D-23-29503R1 

PLOS ONE

Dear Dr. Hermosura, 

I'm pleased to inform you that your manuscript has been deemed suitable for publication in PLOS ONE. Congratulations! Your manuscript is now being handed over to our production team.

Kind regards, 

on behalf of

Dr. Ravi Shankar Yerragonda Reddy 

Academic Editor

PLOS ONE